# Clinical Pharmacy in Psychiatry: Towards Promoting Clinical Expertise in Psychopharmacology

**DOI:** 10.3390/pharmacy9030146

**Published:** 2021-08-21

**Authors:** Hervé Javelot, Clara Gitahy Falcao Faria, Frederik Vandenberghe, Sophie Dizet, Bastien Langrée, Mathilde Le Maout, Céline Straczek, Adeline Egron, Alexis Erb, Guillaume Sujol, Antoine Yrondi, Sébastien Weibel, Philippe D. Vincent, Guillaume Meyer, Coraline Hingray

**Affiliations:** 1Etablissement Public de Santé Alsace Nord, 67170 Brumath, France; Guillaume.Meyer@ch-epsan.fr; 2Laboratoire de Toxicologie et Pharmacologie Neuro Cardiovasculaire, Université de Strasbourg, 67000 Strasbourg, France; 3Institute of Psychiatry, Federal University of Rio de Janeiro (UFRJ), Rio de Janeiro 20211, Brazil; claragitahy@gmail.com; 4Unit of Pharmacogenetics and Clinical Psychopharmacology, Center for Psychiatric Neurosciences, Lausanne University Hospital, University of Lausanne, 1008 Prilly-Lausanne, Switzerland; Frederik.Vandenberghe@chuv.ch; 5Centre de Ressources et d’Expertise en Psychopharmacologie (CREPP) Bourgogne Franche-Comté, 71100 Chalon sur Saône, France; Sophie.Dizet@ch-sevrey.fr; 6Service Pharmacie, CHS de Sevrey, 71100 Chalon sur Saône, France; 7Service Pharmacie, Centre Hospitalier Guillaume Régnier, 35700 Rennes, France; bastienlangree@gmail.com; 8Département de Pharmacie, CHU Henri Mondor, Université Paris Est Créteil (UPEC), AP-HP, 94000 Créteil, France; mathilde.lemaout2@aphp.fr (M.L.M.); celine.straczek@aphp.fr (C.S.); 9Inserm U955 Institut Mondor de Recherche Biomédical, NeuroPsychiatrie Translationnelle, 94000 Créteil, France; 10Service Pharmacie, Centre Hospitalier de Cadillac, 33410 Cadillac, France; Adeline.EGRON@ch-cadillac.fr; 11APF2R, Rouffach Centre Hospitalier, Pôle 8/9, 68250 Rouffach, France; a.erb@ch-rouffach.fr; 12Service Pharmacie, Centre Hospitalier Léon Jean Grégory, 66301 Thuir, France; guillaume.sujol@ch-thuir.fr; 13Service de Psychiatrie et de Psychologie Médicale, CHU de Toulouse, Hôpital Purpan, 31059 Toulouse, France; antoine.yrondi@inserm.fr; 14Centre Expert Dépression Résistante FondaMental, CHU de Toulouse, Hôpital Purpan, 31059 Toulouse, France; 15ToNIC Toulouse NeuroImaging Centre, Université de Toulouse, INSERM, UPS, 31024 Toulouse, France; 16Service de Psychiatrie 2, Hôpitaux Universitaires de Strasbourg, 67000 Strasbourg, France; sebastien.weibel@chru-strasbourg.fr; 17Inserm U1114, 67085 Strasbourg, France; 18Fédération de Médecine Translationnelle de Strasbourg (FMTS), 67000 Strasbourg, France; 19Department of Pharmacy, Institut Universitaire en Santé Mentale de Montréal (IUSMM), 7401 Hochelaga, Montreal, QC H1N 3M5, Canada; philippe.vincent@umontreal.ca; 20Faculty of Pharmacy, Université de Montréal, 2940 Chemin de la Polytechnique, Montreal, QC H3T 1J4, Canada; 21IUSMM Research Center, 7331 Hochelaga, Montreal, QC H1N 3V2, Canada; 22Service Pharmacie, CHU de Strasbourg, 67000 Strasbourg, France; 23Pôle Hospitalo-Universitaire de Psychiatrie d’Adultes du Grand Nancy, Centre Psychothérapique de Nancy, 54520 Laxou, France; c.hingray@chru-nancy.fr; 24CHU de Nancy, Département de Neurologie, 54000 Nancy, France

**Keywords:** clinical pharmacy, mental health, psychiatry, expertise, psychopharmacology

## Abstract

Although clinical pharmacy is a discipline that emerged in the 1960s, the question of precisely how pharmacists can play a role in therapeutic optimization remains unanswered. In the field of mental health, psychiatric pharmacists are increasingly involved in medication reconciliation and therapeutic patient education (or psychoeducation) to improve medication management and enhance medication adherence, respectively. However, psychiatric pharmacists must now assume a growing role in team-based models of care and engage in shared expertise in psychopharmacology in order to truly invest in therapeutic optimization of psychotropics. The increased skills in psychopharmacology and expertise in psychotherapeutic drug monitoring can contribute to future strengthening of the partnership between psychiatrists and psychiatric pharmacists. We propose a narrative review of the literature in order to show the relevance of a clinical pharmacist specializing in psychiatry. With this in mind, herein we will address: (i) briefly, the areas considered the basis of the deployment of clinical pharmacy in mental health, with medication reconciliation, therapeutic education of the patient, as well as the growing involvement of clinical pharmacists in the multidisciplinary reflection on pharmacotherapeutic decisions; (ii) in more depth, we present data concerning the use of therapeutic drug monitoring and shared expertise in psychopharmacology between psychiatric pharmacists and psychiatrists. These last two points are currently in full development in France through the deployment of Resource and Expertise Centers in PsychoPharmacology (CREPP in French).

## 1. Introduction

Clinical Pharmacy is generally considered to have originated in the USA in the 1960s, while the supervising associations such as the American College of Clinical Pharmacy (ACCP) and the European Society of Clinical Pharmacy (ESCP) were created in 1979 [1,2,3]. English speaking countries, and the USA in particular, continue to serve as a benchmark in advancing clinical pharmacy [4]. One of the strongest axes of clinical pharmacy in English speaking countries is based on its specialization and the vision for the future expressed by the ACCP is that “most clinical pharmacy practitioners will be board certified specialists” [5]. The American Pharmaceutical Association created the Board of Pharmaceutical Specialties (BPS) in 1976 in order to create certification to qualified pharmacists, notably in pharmacotherapy (since 1988) and psychiatric pharmacy (since 1992) [5]. Interestingly, it appears that the pharmacist’s qualification as a “drug specialist” may be called into question in the context of the practice of clinical pharmacy in some non-English-speaking countries [4]. The doctor of pharmacy degree (PharmD) may not be sufficient on its own, and may even require a "board-certified pharmacotherapy specialization". Board certification in pharmacy, beyond various elements, improves the interest of the professional for its exercise (increase acceptance by other healthcare professionals, feelings of self-worth, job promotion and salary) and incites the creation of pharmacists able to integrate collaborative drug therapy management with physicians [5].

Twenty-five years ago, the ESCP questioned how clinical pharmacists could play a role in optimizing pharmacotherapy [6]. Three scenarios have been proposed for describing the position of the clinical pharmacist and named "clerk", "controller" and "care manager". Only the last one referred to a proactive model, centered on the patient with a real partnership with physicians, and appeared to be the most desirable scenario for the future of clinical pharmacy [6].

Nevertheless, including in English-speaking countries such as the USA, the interventions of pharmacists were revealed to be formalized under the legal framework within the “collaborative practice agreement” (CPA) [3,5,7]. The CPA defines the relationship between clinical pharmacists and physicians through the initiative “Collaborative Drug Therapy Management” (CDTM) and then authorizes the realization of the “comprehensive medication management” (CMM, which determines whether medications are appropriate, effective and safe) (the previous form, “medication therapy management” (MTM), did not require a CPA) [7]. These elements seem to bear witness to the fact that collaboration between pharmacists and physicians does not appear to be simple or spontaneous.

However, in psychiatry, it clearly appears that knowledge in psychopharmacology is paramount to the quality of clinical practice and thus, pharmacists could be privileged interlocutors for forging a link between pharmacological knowledge and clinical practice. This connection between clinical pharmacology and clinical pharmacy had been thought of by the pioneers of this discipline—Paul Parker (pharmacist) and Charles Walton (pharmacologist)—and was still later relayed by another father of the discipline, Russell R Miller [2].

While recent publications continue to demonstrate the interest and scalability of clinical pharmacy in psychiatry [4,8,9,10], its deployment continues to be held back for various reasons [3,4,11,12].

We propose to carry out below a narrative review of the literature investigating different themes in order to show how clinical pharmacy specializing in psychiatry can respond: (i) to the conventional goals which the discipline sets for itself today (medication reconciliation, therapeutic education); (ii) to new ambitions more oriented towards pharmacotherapeutic optimization and oriented towards privileged collaboration between the psychiatrist (through therapeutic monitoring and shared expertise in psychopharmacology in general), the pharmacist and the patient. These latter elements correspond to an ongoing development in clinical pharmacy specializing in psychiatry in France and is currently in full development in this country through the deployment of Resource and Expertise Centers in PsychoPharmacology (CREPP in French). This integrative model of operation is described at the end of this review.

## 2. Method

In order to limit bibliographic occurrences, the writers of each section (see below) were asked to select the most relevant references based on the following described aims. The aim for parts 3.1 to 3.3 was to describe in a succinct manner the areas considered the basis of the deployment of clinical pharmacy in psychiatry with medication reconciliation, therapeutic education of the patient and the growing involvement of clinical pharmacists in the multidisciplinary reflection on pharmacotherapeutic decisions. The aim for parts 3.4 and 3.5 was to discuss in more depth the data concerning the use of therapeutic drug monitoring and shared expertise in psychopharmacology between psychiatric pharmacists and psychiatrists.

To construct this narrative review of the literature, we have searched on the electronic database PubMed from its conception until 15 April 2021. Firstly, we used the following keywords: “clinical pharmacy” OR pharmacy OR pharmacist AND psychiatry [title/abstract]. This review was used for the following sections: 3.1. Medication reconciliation process in psychiatry; 3.2. Psychoeducation and therapeutic patient education to enhance medication adherence; 3.3. Team-based models of care must include pharmacists and 3.4. Shared expertise in clinical psychopharmacology between psychiatric pharmacist and psychiatrist: why and how? The relevance of the references to be kept was assessed by the authors of each section: parts 3.1 and 3.2 by S.D., M.L.M., C.S., A.G., part 3.3 by B.L. and F.V. and part 3.5 by H.J., C.G.F.F., A.E. and G.S. Secondly, a combination of keywords was associated: “therapeutic drug monitoring” AND psychiatry [title/abstract] for the part 3.4. Therapeutic drug monitoring in psychiatry and implications for pharmacotherapy decision-making. The relevance of the references to be kept in this part was assessed by F.V. and B.L. Extraction was validated by A.Y. and S.W. for parts 3.3 and 3.4, P.D.V and G.M. for parts 3.1 and 3.2 and C.H for part 3.5.

## 3. Narrative Literature Review

### 3.1. Medication Reconciliation Process in Psychiatry

Medication reconciliation (MedRec) consists of collecting all prescribed and non-prescribed drugs at the patients’ admission in order to identify potential discrepancies with the hospital prescription. This process also occurs at discharge to explain all therapeutic changes to primary care providers. The aim is to secure medication management and to improve links between daily care providers and hospitals. MedRec is particularly interesting in psychiatry because sometimes the patient profile makes it difficult to get an exhaustive medication list due to numerous reasons (therapeutic break, cognitive impairment, prevalence of comorbidities) [13]. 

Several studies showed that MedRec reduces adverse drug events and re-hospitalization rate in general hospitals [14]. Fewer studies are available about MedRec in mental health hospitals [15,16]. Nevertheless, a multicenter study showed an error rate of 6.3% in mental health hospitals’ prescriptions, of which more than half were considered as clinically relevant errors [17]. According to local experimentations, performing MedRec at admission enables detection of medication errors for half of patients [18,19] and recognition of new clinically significant drug–drug interactions for more than a quarter of prescriptions [20]. Moreover, MedRec has also had an impact at discharge: an error rate of more than 20% on discharge prescriptions were found in three mental health trusts [21]. In addition, this process could improve the drug follow-up by primary care providers by specifying monitoring parameters (e.g., lithium, clozapine, prescription duration of benzodiazepines). These studies therefore demonstrate that MedRec reduces prescription errors in transition of care and can optimize medication management. 

To complete this process, CMM is being developed in the USA. CMM consists of reviewing all patients’ medications to determine adherence, effectiveness, relevance and safety. This pharmacist-led practice has been demonstrated to improve patients’ treatment goals and to reduce costs [11]. These practices show that pharmacists’ expertise in pharmacotherapy is essential to enhance medication-related outcomes for patients with psychiatric disorders. In turn, the skills of clinical pharmacists in psychopharmacology are often a crucial element to ensure real benefit to the process of MedRec, alone or integrated in CMM, in psychiatry. 

### 3.2. Psychoeducation and Therapeutic Patient Education to Enhance Medication Adherence

Medication non-adherence is broadly observed in psychiatry. Major reviews suggest non-adherence rates between 25 and 90% (depending on type of mental illness and evaluation method of adherence) and consider that at least one half of patients will stop taking their medication at some point in their lives [22,23,24]. A recent meta-analysis showed that the main factors associated with medication non-adherence are patients’ negative attitude toward their medication, lack of insight, negative health beliefs and perceived stigma [24]. Furthermore, consequences of medication non-adherence are numerous, such as treatment resistance, re-hospitalization, risk of self-harm, social disruption and society cost [25]. Psychoeducation and therapeutic patient education (TPE) are continuous processes destined to improve patients’ skills so they can live better daily with their mental illness. The aim is the empowerment of the patient. TPE is more recent and better structured than psychoeducation: it is based on defined programs and led by a trained multidisciplinary team. Few randomized controlled trials (RCTs) on the impact of psychoeducation or TPE are available in the literature. However, a systematic review of RCTs revealed the efficacy of psychoeducation for preventing relapse and improving medication adherence in bipolar disorder [26]. In addition, a RCT reported that a motivational-interviewing-based adherence therapy significantly improved relevant parameters when compared with usual care alone: symptom severity, insight into illness/treatment, functioning, duration of re-hospitalizations and medication adherence over 18 months of follow-up [27].

As pharmacists are directly involved in the dispensing of medicines, they are in a good position to collaborate with patients and support their treatment and assess and promote the importance of medication adherence. A RCT with a 6-month follow-up emphasizes the important role of pharmacists in providing direct patient care in regular pharmacy practice to improve adherence to medications and other patient-reported outcomes [28]. The pharmacist plays a significant role as a collaborator in a team delivering a therapeutic education program. Indeed, a prospective clinical open trial showed that the implementation of a multi-dimensional and inter-sectoral program, including regular pharmacist intervention, enhanced the patients’ adherence significantly up to three months after discharge [29]. As TPE and psychoeducation are now part of national recommendations for treatment and management of mental illness [30,31,32], including pharmacists in these programs seems efficient to improve global patients’ care. Despite the evidence of the relevance of the pharmacist’s role in improving medication adherence, pharmacist’s interventions to promote adherence are not yet common practice. This activity certainly deserves to be further developed and studied. Once again, the clinical pharmacist skills in psychopharmacology appear to be essential so that they can transmit relevant information during therapeutic education to patients suffering from mental disorders. This issue seems all the more important since with the internet, patients have access to a multitude of information, potentially false or simply poorly understood. Thus, in this context, it is important the up-to-date knowledge of the psychopharmacology literature provided by the psychiatric pharmacist can give an objective look.

### 3.3. Team-Based Models of Care Must Include Pharmacists

Team-based models, also referred to as Coordinated Specialty Care, Integrated Collaborative Care or Interdisciplinary Team Care Interventions, were often presented as cost-effective and efficient on key clinical outcomes, especially on re-hospitalization rate [33,34,35,36]. Unfortunately, pharmacists were sparsely included in such settings. Nevertheless, when it happens, pharmacist intervention is found efficient. For instance, pharmacist-convened Multidisciplinary Clinical Team Meetings showed decreased dosages and increased proper use of benzodiazepines [37]. Without really being a team-based approach, another study underlined the positive impact of physician/pharmacist collaboration in reducing polypharmacy in schizophrenia [38]. Clinical recommendations reflect the lack of precise reference to the pharmacist’s role in literature. Based on a recent update of a Cochrane review on Intensive Care Management (ICM), the Royal Australian and New Zealand College of psychiatrists recommend a team based clinical setting to manage schizophrenia and related disorders [39,40]. The 15th statement of the recent update of the American Psychiatric Association recommends Coordinated Specialty Care programs to manage schizophrenia, and especially the first episode of psychosis [41]. 

Team-based care, including pharmacists, has to be the future gold standard of psychiatric care as complex treatments are being developed. The new antidepressant esketamine is an example, where the collaboration of the prescriber with the pharmacist is necessary to ensure that treatment-resistant depression is correctly defined (check if former treatments were used at sufficient dose and duration, to be sure that another strategy is not possible). The cost of this treatment is unusual in psychiatry. Therefore, while waiting for an evolution of the funding model in psychiatry, team-based management allows for reinforcement of the proper use of those complex treatments.

In mental healthcare, treatment resistance is a challenge for the management of depression as well as of schizophrenia. In fact, treatment resistance is associated with higher hospitalization rates and emergency department visits, longer hospital stays and poorer prognosis [42]. It is not always easy to distinguish real resistance to other potential causes of persistent symptoms. Firstly, although pharmacotherapy is essential for treating most psychiatric disorders, its success is often limited by medication adverse effects: a French report published in 2011 showed psychotropic drugs were responsible for 16% of reported adverse effects [43]. These adverse effects, which often go under-recognized by both patients and physicians, have as a consequence lack of adherence. Secondly, as for antipsychotics, medication underexposure due to not achieved dosage or pharmacokinetic factors can lead to subtherapeutic plasma levels [44]. Thirdly, many drugs (e.g., stimulants, anabolic steroids, angiotensin converting enzyme inhibitors, anticholinergics, tricyclic antidepressants, antiepileptics, benzodiazepines, beta-blockers or dopamine receptor agonists) can cause psychiatric symptoms (depression, anxiety, mania or psychotic symptoms) which can be confused with psychiatric diagnoses. Therefore, for patients with new onset or treatment-resistant psychiatric symptoms, iatrogenicity should be considered to avoid confusion with a sign of psychiatric illness [45]. Concerning schizophrenia, antipsychotic-induced dopamine supersensitivity psychosis or chronic low-grade peripheral inflammation can lead to treatment resistance [46,47]. 

Beyond the knowledge required in psychopharmacology to carry out MedRec or TPE/psychoeducation, the integration of the pharmacist into the medical team allows the psychiatric pharmacist to participate directly in the pharmacotherapeutic optimization in partnership with the psychiatrists. Detecting treatment resistance and analyzing their causes are essential to improve patient care. The involvement of the psychiatric pharmacist from MedRec to team consultation in cases of resistance helps to give the most suitable treatments to the patient at each stage of their care.

### 3.4. Therapeutic Drug Monitoring in Psychiatry and Implications for Pharmacotherapy Decision-Making

Therapeutic drug monitoring (TDM), defined as the quantification of drugs in serum or plasma samples, is a tool to enhance treatment response and/or reduce dose-related side effects. Historically, TDM was used in psychiatry to enhance the treatment safety of drugs known to have a narrow therapeutic range, such as tricyclics or lithium [48]. With the development of safer psychopharmacologic agents (at least for acute conditions), such as selective serotonin reuptake inhibitors (SSRIs), TDM is frequently used to personalize treatment with the aim of enhancing the therapeutic response. In this section, we will briefly overview the basic principles of TDM for psychiatry and discuss common clinical situations where the information provided by this tool can aid pharmacotherapy decision-making. 

For all the information that we provide below, the psychiatric pharmacist trained in the use of TDM can guide the psychiatrist on the opportune moments to carry out the assays, to then analyze the results obtained and know how to adapt the treatment later. We transmit hereafter concrete elements showing how this expertise can be valuable to psychiatrists in the context of a collaboration with the psychiatric pharmacists.

Although TDM has become more readily available, it is only informative for certain psychotropic drugs and in specific conditions. For example, agomelatine, an antidepressant with melatoninergic properties, has a short half-life (one to two hours) and therefore its concentration in standard conditions will not be detectable. The Arbeitsgemeinschaft für Neuropsychopharmakologie und Pharmakopsychiatrie (AGNP) taskforce provides recommended levels for many common psychiatric drugs [49]. Briefly, the recommendations are divided into four grades: strongly recommended (e.g., lithium), recommended (e.g., escitalopram), useful (e.g., memantine) and potentially useful (e.g., lorazepam). The grades are not only based on the existence of a correlation between blood concentration and pharmacologic effects, but also on the evidence of reported therapeutic reference values. In addition to the recommended level for each compound, two major principles should be respected to properly compare obtained measures with reference values. First, steady-state conditions, defined as an equilibrium between medication absorption and elimination, should be achieved. In practice, steady-state conditions are reached after 4-5 half-lives (approximately one week for SSRIs with an elimination half-life of 30 h, but several months for a long-acting antipsychotic with an apparent terminal elimination half-life of 30 days). Secondly, blood samples should be drawn to obtain trough concentrations (the minimal concentration, just before oral drug intake). In practice, blood sampling may be performed in the morning, just before drug intake (or just before the next injection of a long-acting antipsychotic) and independently of the daily dosage regimen. In specific situations (e.g., a short half-life compound given once daily at bedtime), trough concentrations should be extrapolated to ensure a proper comparison with reference values. Appropriate interpretation and communication of the results are also of major importance. TDM results should not only include drug measured concentration, but also concentration to dose ratios, and if evidence is available, relevant active metabolite concentrations and parent: metabolite ratios. Additionally, therapeutic reference ranges and a clinical interpretation should be given in the final report. The latter requires additional information such as co-prescribed agents, renal function, weight, therapeutic response, and involves a team of trained psychiatric pharmacists and psychopharmacologists. 

In everyday clinical practice, TDM is a valuable tool for adjusting psychotropic dosage in response to an insufficient treatment response within established therapeutic dose ranges. In this situation, TDM can help the prescriber to distinguish between three situations: An uncertain treatment adherence;Altered metabolic pathways involved in the drug metabolism;A treatment resistance.

The use of TDM for measuring adherence needs to be interpreted with caution (e.g., short half-life molecules will only be informative for adherence one to two days before blood sampling). If non-adherence is confirmed and a good response is previously documented, it can help prescribers to reintroduce the treatment with measures to enhance adherence, or switch to a long-acting formulation. Non-optimal plasma levels in compliant patients treated at therapeutic dose ranges can also be partly explained by genetic factors affecting drug pharmacokinetics. Cytochrome P450 enzymes (CYP450), which have important genetic variability, metabolize a large number of psychotropic drugs. The relation between these genetic variants and clinical outcome has been investigated in two large-scale retrospective studies which reported higher rates of therapeutic failure for escitalopram and risperidone within one year, in patients carrying specific CYP2C19 and CYP2D6 genotypes [50,51]. Unfortunately, to date, pharmacogenomic data are only able to capture a minority of CYP450 variations (except for CYP2D6, CYP2C9 and CYP2C19 isoforms) and therefore are not informative enough for a broad clinical use. TDM will guide prescribers to define an adequate dose titration if the treatment is well-tolerated, or to switch to a drug metabolized by another pathway or non-metabolized. Finally, treatment resistance, defined as an inadequate response with an adequate treatment duration and adequate serum level, is a common issue in psychiatry [52] . Once this situation is confirmed by obtaining blood levels close to the upper value of the therapeutic reference range, the prescriber should consider changing the treatment according to medication history or investigate augmentation strategies.

TDM is also informative in specific clinical situations where drug pharmacokinetics can be suddenly and/or temporally altered, such as in the presence of drug–drug interaction, in the case of bariatric intervention or in pregnancy. As previously mentioned, several psychotropic drugs are metabolized by CYP450 and thus are at risk of pharmacokinetic alterations due to environmental substances and/or medications with inducing or inhibiting properties. However, the magnitude of dosage adaptation in this situation is difficult to predict since only a few drug–drug interactions have been prospectively studied, and most of them only in small healthy cohorts and at specific dosages. For example, the manufacturers of paliperidone mention that the adjunction of 400 mg/d of carbamazepine will lead to a 37% decrease in the area under the curve. Since this is not considered to be clinically significant, no dosage adjustment is needed [53]. However, independent data found that paliperidone concentrations decreased in a dose-dependent manner between 200 and 600 mg/d of carbamazepine, suggesting that the effect is more important than initially reported by the manufacturer [54]. Another common example is the dosage adjustment of clozapine in a context of smoking cessation. To compensate for the decrease in CYP1A2 activity and thus avoid potential intoxication, recommendations suggest reducing clozapine dosage by 30% before initiating a smoking cessation process [55]. However, due to an important intervariability of CYP1A2 induction (estimated as up to 7.3-fold) and the gradual process of induction or return to basal activity (three to four weeks), monitoring of clozapine blood levels should be used to optimize dosage titration in addition to an initial dose reduction of 30% [56]. Psychiatrists are increasingly confronted by patients who are eligible for, or have a past medical history of, bariatric intervention, given the high prevalence of obesity in this population [57]. Although bariatric surgery appears to have a positive effect on depression during the first few months (the so-called "honeymoon period"), symptoms tend to increase 36 months after the intervention [58]. This observation suggests that, in addition to adherence issues due to the amelioration of symptomatology, there is also the possibility of drug malabsorption. Several prospective case-series studies and retrospective case-control studies, mostly including patients treated with antidepressants and with Roux-en-Y gastric bypass intervention, reported important changes in concentration-dose ratio shortly after surgery [59]. These absorption changes are difficult to predict and thus an individualized approach should be used to optimize pharmacotherapy. In addition to a dose modification, other therapeutic decisions can be considered if TDM indicates a malabsorption, such as dividing the dosage into several daily drug intakes, switching to an injectable form, (e.g., from oral antipsychotics to long-acting antipsychotics) or crushing tablets (according to the package insert) or use of liquid or oral dissolving formulations in some cases. Due to several physiological changes during pregnancy, pharmacokinetic alterations of many drugs are expected which can have a greater or lesser clinical impact. A comparison of dose-adjusted antidepressant concentrations in the third trimester versus the baseline period found several changes, some probably clinically non-significant (sertraline), but some clinically concerning (citalopram) [60]. The major limitation is the scarcity of the data from which to draw recommendations, except for certain mood stabilizers such as lithium, for which specific recommendations are published [61]. To optimize decision-making in these three situations, TDM values should be compared (if it is possible), with basal values obtained before the introduction of an interaction drug, bariatric surgery or in the early stages of pregnancy.

Although precision medicine integrating genetic and other individual factors has become more widespread in psychiatry, decision-making in psychopharmacology still poses a challenge since up to 50% of depressed patients do not respond to the first-line treatment choice [62]. These discussed common clinical situations highlight that TDM, associated with clinical evaluation, is a relevant tool for evidence-based pharmacotherapy optimization, such as dose titration, modification of formulation or treatment switch. However, implementation of TDM in clinical routines is a challenging process due to the specific conditions of blood sampling, the need for additional prescription information and by the clinical interpretation of these results. These issues can be overcome with adequate training of psychiatric pharmacists who can then guide psychiatrists in the rational use of TDM of psychotropic drugs.

### 3.5. Shared Expertise in Clinical Psychopharmacology between Psychiatric Pharmacist and Psychiatrist: Why and How?

Training in psychopharmacology is necessary for providing high quality psychiatric care and the intervention of the psychiatric pharmacist is relevant in this regard [63,64,65]. In a study in three psychiatric hospitals, psychotropic polypharmacy was more frequent in hospitals without psychopharmacology units; access to clinical psychopharmacology expertise and teaching may be an important factor for promoting good prescribing practices [66]. Supporting expertise in clinical psychopharmacology appears to be of great importance, but the question is not why, but remains how and by whom this necessary expertise is to be provided [67].

Furthermore, simplistic categorical diagnoses are abundantly used in contemporary psychiatric practice and have contributed to pharmacotherapy choices dictated by guidelines/algorithms [4,68]. Therefore, this decline in clinical curiosity in favour of a more standardized approach is potentially associated with impoverishing clinicians’ knowledge of the pharmacological tools available [4,68]. 

We believe that the acquisition of refined and specialized knowledge by psychiatric pharmacists is the paramount to real efficiency of clinical pharmacy in mental health. Thus, participation in therapeutic optimization in psychiatry by psychiatric pharmacists can only be conceived through the development of sufficient expertise in clinical psychopharmacology. From our point of view, MedRec and the other quality frameworks provided by pharmacists who have become "care managers’’ partly forget the initial training in clinical psychopharmacology which must enable the psychiatric pharmacist to deliver information that is relevant and compatible with clinical reality.

For several years now, we have created in France a local skills network specializing in psychopharmacology [4,69]. Recent data bear witness to the fact that this network of expertise, involving both psychiatric pharmacists and psychiatrists, receives a large and growing number of requests, with a high level of satisfaction expressed by the doctors requesting it [4]. This system is also proving effective in terms of publications and in particular for formalizing professional recommendations [4]. The expertise is particularly offered during the beginning of care in order to optimize the first-line psychopharmacotherapeutic strategy and also at a later stage for treatment modification in the event of treatment resistance or adverse effects [4]. 

Our network operating model in psychopharmacology is an advanced form of “case conferencing intervention” as observed in Australian and Scandinavian settings [12,70]. In our experience, the shared decision-making model about pharmacological optimization is particularly well perceived and desired by young psychiatrists. In this new collaboration report, the transformation of drug compliance into adherence and concordance has changed the conception of young practitioners who seek more of a joint final decision and shared responsibility for psychopharmacotherapy [22].

The shared medical decision is built beforehand by sharing clinical information and deliberation. In the light of this information, the psychiatric pharmacist can bring the most recent data from the literature and reflect with the psychiatrist on the best way to translate the theoretical information into an individualized treatment plan based on the patient’s history [4]. In this perspective, the role of the psychiatric pharmacist is also interesting for making it possible to design a therapeutic decision which takes into account from the beginning the drug–drug and drug–disease interactions [4].

Psychiatrists are aware of recommendations or guideline algorithms, but in some countries it is not always possible to apply them, often because they are incompatible with real-life conditions. Therefore, because it is difficult for psychiatrists in some settings to follow the guidelines and find out how they can be adapted to the treatment, the psychopharmacological expertise consists of conciliating with prescribers the algorithmic thinking from recommendations to an adapted view for a specific clinical situation. Nevertheless, a good understanding of the psychiatrist–psychiatric pharmacist pair from this perspective can only be conceived if the pharmacist has a strong clinical experience in psychiatric settings (not solely as a punctual "clinical pharmacist" in mental health).

Although evidence- or consensus-based guidelines are built on data from clinical trials, meta-analyses, pharmacoepidemiologic and objective data about efficacy and safety, the use of pharmacodynamic data to make these recommendations applicable is essential. While data from large studies or meta-analyses is often important in shaping recommendations, they do not necessarily replace them. In the Star*D study, a large naturalistic study on treatment options for depression and resistant depression, the results highlighted the relevance of using the combination venlafaxine + mirtazapine [71]. This combination, sometimes referred to as “California Rocket Fuel”, appeared in this study to be comparable in terms of efficacy to an MAOI (tranylcypromine), with better tolerance [71,72]. Due to a certain pharmacological complementarity, this combination of antidepressants has since established itself as a relevant strategy in the event of failure of multiple antidepressant monotherapy schemes. It should nevertheless be noted that this strategy is based both on a low level of evidence and that contradictory results are reported [73,74]. In addition, these data can be interpreted as an incentive to go quickly to combinations of antidepressants by forgetting for example that in the event of failure of a first SSRI the response rate to a second option falls to 50% and that for these same treatments, dosage optimization may make sense before validating a genuine situation of resistance [75,76]. Optimizing the use of antidepressants takes on even more significance with the use of venlafaxine for which the emergence of the norepinephrine reuptake inhibitor effect is only observed with increasing doses [77]. Thus, using venlafaxine at a dose ≤ 150 mg / day may be equivalent to testing an SSRI and may therefore appear in the event of partial efficacy and good tolerance to an incomplete therapeutic trial.

In a recent meta-analysis [78], integrating 21 antidepressants with two tricyclics, the authors were surprised to find that while amitriptyline seems to present itself as the gold standard in terms of efficacy, clomipramine is less efficacious than most of the other antidepressants reviewed. However, these data must be analyzed considering the following limitations: (i) they do not present any direct comparison data between the two tricyclics, (ii) RCTs included are about five times higher for amitriptyline (96) than for clomipramine (20). Furthermore, it should be noted that old comparison data between the two tricyclics suggested a delayed response to clomipramine [79]. Clomipramine also presents a potentially higher risk of adverse effects which may also induce a slower dose inflation due to their regularly transient nature [79,80]. Taken together, these elements may explain that 8 weeks evaluation favors amitriptyline over clomipramine without demonstrating, in the absence of direct comparison over a sufficient period, the certainty of a superiority of one agent over another. Here again, it therefore appears that beyond a specific agent within an antidepressant class, the importance of therapeutic optimization will play a decisive role.

Alongside this good understanding and application of data from the literature, it is also necessary to make the guidelines suitable by adapting them to the specificities of the patient: all of the other treatments in progress, all of the comorbidities, and psychotropic treatments previously tried in the current indication with their efficacy and tolerance levels. The empirical approaches sustained by guidelines should never forget that it is from the individual story of each subject that the recommendations are constructed for the most specific and relevant care [81].

Furthermore, from our point of view, the correct application of the guidelines should also include at each stage the possibility of relying on TDM. This strategy could be considered for treatments benefiting from a high level of recommendation according to the AGNP task force (two first grades: strongly recommended and recommended) in order to exclude partly pharmacokinetic resistance and to increase the level of certainty about the existence of pharmacodynamic resistance.

### 3.6. Global Collaborative View of Psychiatric Pharmacists and Psychiatrists

This narrative review of the literature makes it possible to formalize an extended vision of the field of collaboration between psychiatric pharmacists and psychiatrists in patient care. The interventions of psychiatric pharmacists may be broken down into two levels. A first level integrating conventional clinical pharmacist missions such as MedRec—integrated or not in CMM or MTM—and TPE. The second level is based on the principle of Interdisciplinary Team Care Interventions and corresponds in our practice to an advanced form of CCI, offered by some countries. This level revolves around a network of practitioners, with psychiatric pharmacists and psychiatrists recognized as experts on a (or more) targeted domain(s), in order to formalize guidelines tailored to the particularities of each patient and offer advice about the rational use of TDM for optimize the prescription of psychotropics. Psychopharmacotherapeutic optimization co-developed by psychiatric pharmacist and psychiatrists and integrated: MedRec, advanced CCI and TPE is the challenge proposed by the new Resource and Expertise Centers in PsychoPharmacology (CREPP—Centres de Ressources et d’Expertise en Psychopharmacology—in French) [4].

We propose a synthetic vision of the collaborative view between psychiatric pharmacists and psychiatrists to promote shared expertise in clinical psychopharmacology through development of CREPPs on Figure 1.

## 4. Conclusions

In several countries, and in particular non-English-speaking countries, such as France, the pharmacist is often more of a quality manager than a true clinical pharmacist. To overcome this “identity crisis” it is undoubtedly necessary to question the visions that pharmacists have of their own aptitudes and wishes in relation to what clinical practice is [3,82,83]. This perspective also places them in front of a potential socio-professional transformation leading them to transition to a conception of care closer to that of physicians [82,83].

We believe that MedRec and TPE are important bases to keep as the first foundations in the construction of clinical pharmacy in mental health. Nevertheless, psychiatric pharmacists must now assume a growing role in team-based models of care, as well as to participate intensively in the shared expertise in psychopharmacology aiming for therapeutic optimization and a shared decision on treatments. In this perspective, increasing skills in psychopharmacology and developing expertise in TDM in particular, should build the future legitimacy of the working partnership between psychiatrists and psychiatric pharmacists. This model of shared expertise in psychopharmacology is currently in full expansion in France through the deployment of Resource and Expertise Centers in PsychoPharmacology [4].

## Figures and Tables

**Figure 1 pharmacy-09-00146-f001:**
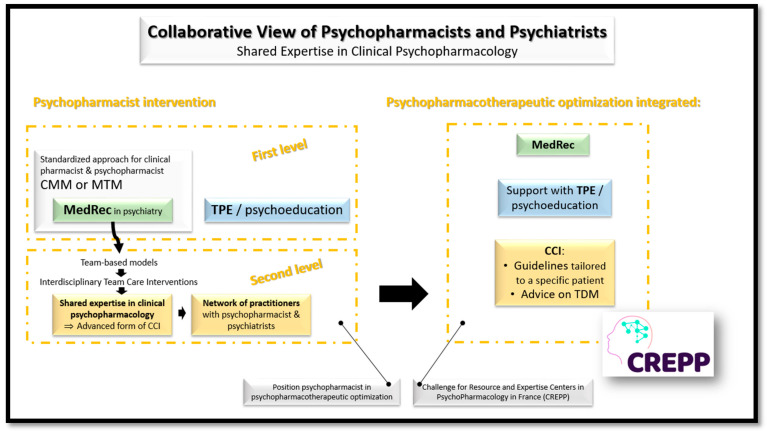
Collaborative view of psychiatric pharmacists and psychiatrists for a shared expertise in clinical psychopharmacology and the development of CREPPs. CCI: Case Conferencing Intervention—mainly developed in Scandinavian countries and in Australia; CMM: Comprehensive Medication Management and MTM: Medication Therapy Management—mainly structured in English speaking countries; CREPPs: Centres de Ressources et d’Expertise en PsychoPharmacologie = Resource and Expertise Centers in PsychoPharmacology—currently in full expansion in France; MedRec: Medication reconciliation—widely developed around the world; TDM: Therapeutic Drug Monitoring—applied to psychotropic drugs has been particularly developed in certain German-speaking countries—Germany, Austria and Switzerland—which have proposed the "Consensus Guidelines for Therapeutic Drug Monitoring in Neuropsychopharmacology" at the initiative of the TDM task force of the Arbeitsgemeinschaft für Neuropsychopharmakologie und Pharmakopsychiatrie (AGNP); TPE: Therapeutic Patient Education—widely developed around the world.

## Data Availability

Not applicable.

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
