# Peer review of "Clinical Pharmacy in Psychiatry: Towards Promoting Clinical Expertise in Psychopharmacology"

_pharmacy, 2021, doi:10.3390/pharmacy9030146_

Round 1

Reviewer 1 Report

I applaud what the authors are trying to do with this manuscript to set a framework for progression and development of the role of the psychiatric pharmacist in France. I do wonder if there are systemic factors at play that have prevented progression and development of the clinical pharmacist role compared to pharmacist professional activity advancement in North American/English speaking countries and whether these remain to be broken down in order to move to the newly proposed paradigm. 

A reasonable organization of the paper is made in terms of the major aspects of the clinical pharmacists role (what about education of nursing and psychiatric staff and trainees ? I see no mention of this and this is another way to gain recognition for pharmacists' competence in psychopharmacology and over times leads to these trainees to increasing invite pharmacists collaboration in medication decision making. 

Some aspects that need to change - would really prefer an alternate term to "Anglo-Saxon". According to wikipedia: The Anglo-Saxons were a cultural group who inhabited England in the Early Middle Ages. I don't feel this  accurately describes non-US English speaking countries.

There are some statements in the paper re TDM, pharmacogenomics, CYP enzymes and pharmacokinetics that unfortunately illustrate a lack of depth of knowledge of these topics to knowledgeable readers in some cases and should be revised. See the marked up pdf I have provided to the editor with several additional comments. I would completely remove the statements and reference to trastuzumab intrathecal injections for treatment resistant schizophrenia. This 2004 paper was entirely theoretical, and there have been no subsequent case reports, so the field is not 'upcoming'.

Recommend English language review - there are several phrases where wording is awkward or not quite right and could be improved. (That said, it came out better than if I was to try to write a manuscript in French!).

Need to decide if you're going to call the role the psychiatric pharmacist (my preference) or psychopharmacist and use that term consistently throughout the manuscript.

Can standardize the number - since the sections are numbered 1.3, 1.4 etc should refer to the sections the same way in section 1.1

Some citations have some formatting issues, or need access dates for URLs

I am one of 2 adolescent psychiatric pharmacists in my province, but one thing we have recently been providing is pharmacotherapy support to pediatricians and family doctors around our province via zoom/Skype for business. The recent pandemic has accelerated this delivery, but if there are relatively few psychiatric pharmacy experts in the country at CREPP, you might want to consider how the expertise of a center such as CREPP can be delivered to providers around the country by remote consultation.

Finally - you don't mention payment/compensation structures at all - not sure if this was on purpose, but seems like there is a cost consideration for the training and development of such expertise, and just wondered how you would view the financial aspects of such a paradigm shift - perhaps this is for another paper. 

Is pharmacist prescribing an available tool/option for you? It is not yet in our province, but this opens up other opportunities if it is available, and may be worth theorizing or mentioning what additional services this would allow CREPP to perform if it was to become available.

Author Response

Please see "REVIEWER 1 15072021" PDF

Reviewer 2 Report

In order to limit bibliographic occurrences, the writers of each section (see below) 117 were asked to limit themselves to 15 references for parts 3 to 5 and to 20 references for 118 parts 6 and 7 each. - please clarify

There is a lot of text, I would suggest some drawing concepts to be included, as well as tables for better reading.

Very interesting read and nice work!

Author Response

Please see "REVIEWER 2 15072021" pdf

Reviewer 3 Report

Thank you for inviting me to review this nice paper entitled "Clinical pharmacy in psychiatry: towards promoting clinical expertise in psychopharmacology". This is an interesting review regarding the potential role played by pharmacists in the management of psychopharmacological therapy and in improving the adherence of psychiatric patients to medication. I have some comments that may help the authors improve the quality of the paper.

  • My concerns mainly regards paragraph 1.1, which appears added in a second moment by the authors. I do not think it is necessary to describe in such levels of the details the methodology followed, as it is a narrative review, not a systematic review. Particulrarly the first paragraph of this part appears out of place and poorly informative
  • I would also suggest to re-number the paragraph in 1, 2, 3 etc instead of 1.1., 1.2
  • It would be important to discuss whether there might be cross-cultural differences that may hamper the adaptation of the proposed model of care in different parts of the world.

Author Response

Please see "REVIEWER 3 15072021" pdf

Round 2

Reviewer 1 Report

Thank you for your response letter and making the requested changes. Good luck with implementation of this new paradigm. Just some minor typos to correct (see attached pdf).

Author Response

"Thank you for your response letter and making the requested changes. Good luck with implementation of this new paradigm. Just some minor typos to correct (see attached pdf)."

Thank you very much

See « Academic Editor & reviewer 1 Notes- authors reply (1) »

Reviewer 3 Report

The authors have adequately addressed my comments.

Author Response

"The authors have adequately addressed my comments."

Thank you very much